# LARGE LANGUAGE MODELS DO NOT MAKE COMPLETE USE OF MATH REASONING DATA

## ABSTRACT

In deep learning, increasing dataset size has been shown to improve the performance of deep neural networks. However, it is unclear if these models are able to make complete use of the data that they are trained on. Understanding this is especially important in the current large language model era, where data scarcity has become a pressing issue. We discover that when performing fine-tuning on mathematical reasoning tasks, adding more training data causes the model to incorrectly answer a large portion of previously correctly answered test samples. This remains true even with popular test-time scaling techniques, which can iron out inconsistencies in model predictions. To better understand this phenomenon, we show both empirically and theoretically that models trained using Supervised Fine-Tuning and Reinforcement Learning are incapable of making complete use of the data that they are trained on, where models trained on the same data learn very different functions across different random seeds, exhibiting high predictive multiplicity. This work contains novel insights that can aid in improving a model's ability to effectively scale its performance with more data.

## 1 INTRODUCTION

It is generally understood that increasing training data can improve the performance of deep neural networks (Kaplan et al., 2020; Sorscher et al., 2022). To take advantage of this, an active effort has been applied to procure more data or generate it synthetically, specifically for settings involving large language models. However, many recent studies have shown that we are approaching the limits of human-generated text data (Muennighoff et al., 2023; Tirumala et al., 2023) and suggest that synthetic data may be a promising direction. Recent works have even attempted to study scaling laws of synthetically generated data to confirm this hypothesis, where most studies suggest that current generation techniques produce data that do not match human-generated samples. In an effort to procure more data, however, a crucial factor has been understudied: *Do deep neural networks make complete use of the data they are provided?*

To answer this question, we begin by performing a deeper analysis of how model performance scales with an increase in training data. More specifically, rather than simply observing a net increase in test performance with increasing training data, we study how individual test samples are impacted by increasing data. We note that our scope is defined to cover large language model training on math reasoning tasks with supervised fine-tuning and reinforcement learning, a setting where data scarcity has become a concern. Interestingly, we observe that on addition of more training data, while test performance rises, a large portion (10-15%) of previously correctly answered test samples are now incorrectly answered with more data. This behavior is observed even with test-time scaling techniques such as majority voting, which are used to overcome inconsistencies in models' outputs produced through non-deterministic behaviors and elicit better performances.

To better understand this phenomenon, we investigate incorrectly answered sample groups and perform a fixed set analysis to learn that these models are simply not capable of making complete use of the training data that they are provided. Large language models trained on the same data but with different random seeds can learn vastly different functions, where all models have similar test performances, but the intersection of correctly answered test samples by these models is very small. Observing that large language models exhibit notably high predictive multiplicity, we link our empirical findings to established theory and show that this behavior is attributed to an inability

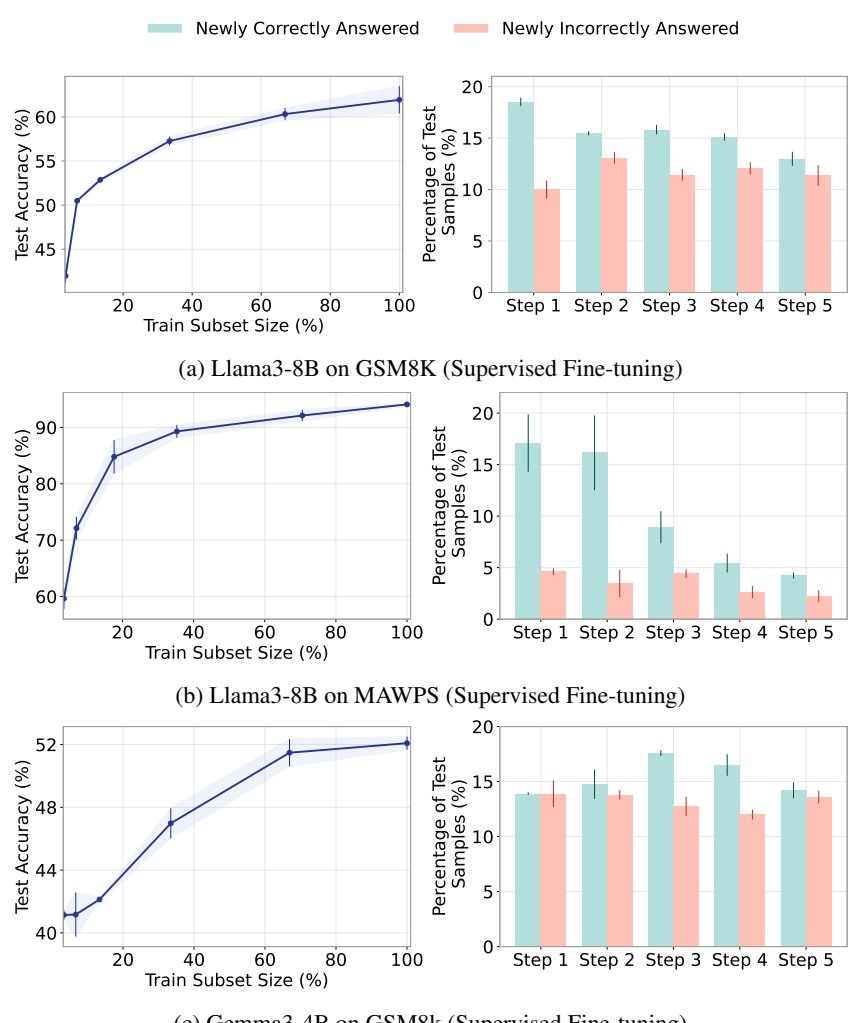

(a) Llama3-8B on GSM8K (Supervised Fine-tuning)

(b) Llama3-8B on MAWPS (Supervised Fine-tuning)

(c) Gemma3-4B on GSM8k (Supervised Fine-tuning)

Figure 1: With additional training data, there exist many previously correctly answered test samples that become incorrectly answered when trained using supervised fine-tuning.

to make full use of the training data provided. We provide an in-depth understanding of this novel failure mode of the current large language model training paradigm.

## 2 BACKGROUND

**Reasoning in Deep Learning.** Recent works aim to understand if deep networks are capable of drawing out underlying rules from training instances. Early works show that deep networks instead rely on approximate rules/heuristics or statistical features to solve reasoning tasks, making them unable to adapt to newer domains that rely on the same set of rules that can correctly solve the original task (Zhang et al., 2023; Liu et al., 2023; Nikankin et al., 2025). However, with the advent of more robust reasoning models, it is becoming increasingly evident that these models are capable of performing reasoning by drawing out rules and learning when to apply them to solve a problem (DeepSeek-AI et al., 2025), but still struggle to perform certain tasks.

**Mathematical Reasoning.** Assessing and understanding the mathematical reasoning ability of deep neural networks has become an important area of study within the field of deep learning (Zhou et al., 2024; Lee et al., 2024). (Lample & Charton, 2020) show that deep networks attain good performances in slightly complex mathematical tasks such as symbolic integration and the solving of differential equations. Following this, Cobbe et al. (2021) showed that heavily parameterized

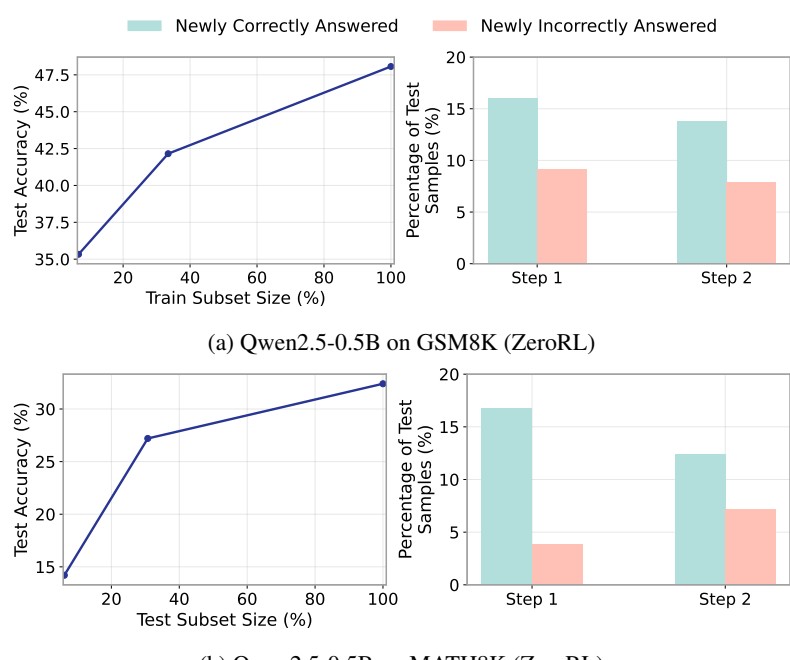

(a) Qwen2.5-0.5B on GSM8K (ZeroRL)

(b) Qwen2.5-0.5B on MATH8K (ZeroRL)

Figure 2: With additional training data, there exist many previously correctly answered test samples that become incorrectly answered when trained using reinforcement learning.

models fail when solving math word problems with simple arithmetic tasks and that attaining good performance generally requires the utilization of additional compute, especially during test time. Hendrycks et al. (2021) further reinforced this claim and showed that in word problems with more complex mathematical operations, deep networks are unable to perform. Today's reasoning models, although less parameterized than the ones discussed in (Cobbe et al., 2021), are better at solving these tasks but still struggle to solve many problems within these tasks, even with test-time scaling.

**Math Reasoning Datasets.**  We consider 3 math reasoning datasets in this work: MAWPS (Koncel-Kedziorski et al., 2016), which comprises simple math word problems, GSM8K (Cobbe et al., 2021) which comprises harder grade school math word problems, and MATH (Hendrycks et al., 2021; Zeng et al., 2025), which comprises much harder high school math word problems. These datasets are representative of most math reasoning tasks studied in current literature.

**Inclusion of Reasoning Steps in Training Data.**  To improve the mathematical reasoning ability of deep networks, most datasets comprise of chain-of-thought reasoning steps, where instead of standard question-answer pairs, solutions often also contain intermediate reasoning steps that eventually arise at the answer. Inclusion of such intermediate reasoning steps has been shown to improve performance significantly. In addition to these, many works propose augmenting or transforming training instances to induce new ways of reasoning about mathematical word problems. Weng et al. (2023); Jiang et al. (2024) introduce backward reasoning questions for training instances, where models are taught to reason about a question better by starting from the answer and working back towards an initially provided value. Yu et al. (2024) aim to augment a training set with rephrased questions that arrive at the same answer, thereby allowing models to draw out similarities between questions that seem different on the surface.

**Scaling Test-time Compute.**  Practitioners have shown that scaling test-time compute helps improve performance significantly on math reasoning tasks. Cobbe et al. (2021) show that by training verifiers and performing ranking of many candidate solutions, deep networks were capable of attaining much better performances on mathematical reasoning benchmarks. Wang et al. (2023) follow a simple approach of sampling many candidate solutions and simply picking the most common answer, also known as majority voting. Others use reward models to score candidate solutions and simply pick answers with the highest score (Snell et al., 2025).

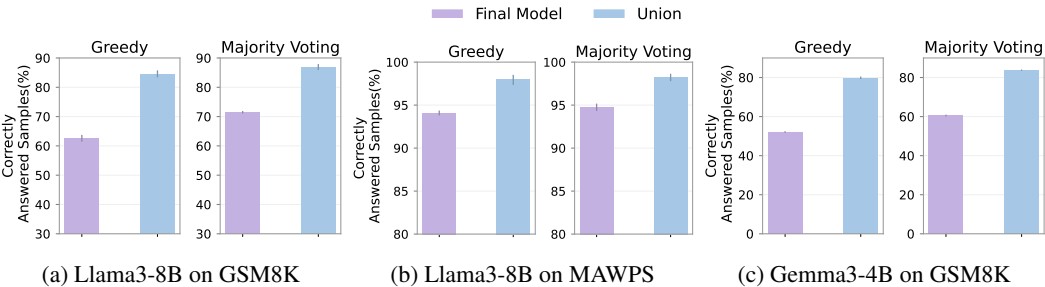

(a) Llama3-8B on GSM8K     (b) Llama3-8B on MAWPS     (c) Gemma3-4B on GSM8K

Figure 3: The union of all correctly classified test samples by models trained on different train subset sizes is significantly greater than the number of correctly classified samples by the model trained on the entire dataset.

**Supervised Fine-tuning (SFT).** Supervised fine-tuning (Wei et al., 2022; Chung et al., 2022) is the de facto training objective used to fit large language models to specific tasks because it is relatively inexpensive and does not need verification. In supervised fine-tuning, the model is trained for a few training steps to mimic the training distribution by learning to predict the next token given prior context. We define the supervised fine-tuning objective as follows:

Given a pretrained causal LM $(p_\theta)$ and a dataset $\mathcal{D} = \{(x_i, y_i)\}_{i=1}^N$ of inputs/prompts $x_i$ and target outputs $y_i = (y_{i,1}, \ldots, y_{i,T_i})$, SFT minimizes the token-level negative log-likelihood (teacher forcing):

$$\mathcal{L}_{\mathrm{SFT}}(\theta) = -\frac{1}{N} \sum_{i=1}^{N} \sum_{t=1}^{T_i} w_{i,t} \, \log p_\theta\big(y_{i,t} \mid x_i, \, y_{i,<t}\big),$$

where $w_{i,t} \in \{0, 1\}$ masks out prompt tokens (and any tokens we choose not to supervise), and $y_{i,<t} = (y_{i,1}, \ldots, y_{i,t-1})$.

Due to its indispensability and general reliability, it is critical to study how supervised fine-tuning makes use of the data it is provided.

**Reinforcement Learning (RL) with verifiable rewards.** Recent work has shown that simple reinforcement learning with verifiable rewards on base language models can elicit strong reasoning behaviors (DeepSeek-AI et al., 2025; Zeng et al., 2025; Gandhi et al., 2025). Large language models are commonly trained with reinforcement learning through Group Relative Policy Optimization (GRPO) (Shao et al., 2024) and thus, it is the technique used for RL training in this paper. More specifically, we perform ZeroRL, which is RL training on the base model without any prior supervised fine-tuning step (Zeng et al., 2025).

**Parameter-Efficient Fine-Tuning.** Parameter-efficient fine-tuning techniques were created to make it more efficient to fit a model to a particular task. More specifically, it freezes the weights of a model and trains a much smaller set of extra parameters to reduce memory requirements. LoRA (Hu et al., 2022), QLoRA (Dettmers et al., 2023), and LoftQ (Li et al., 2024) are the primary PEFT techniques used in practice. In this work, we make use of LoRA and LoftQ.

## 3 STANDARD SCALING LAW ANALYSES OBSCURE LOSS OF IMPORTANT INFORMATION.

We begin our analysis by performing supervised fine-tuning of large language models (Llama3 (Dubey et al., 2024), Gemma3 (Team et al., 2025)) on math reasoning tasks (GSM8K (Cobbe et al., 2021), MAWPS (Koncel-Kedziorski et al., 2016). To understand how additional data improves test-time performance, we fine-tune the model on increasing subsets of the dataset, where every time we increase the subset size, the new subset is always a superset of the previous subset. Note that the subsets are consistent across different random seeds. Consistent with previous analyses, as one scales the size of the train dataset, test-time performance improves, with large initial increases that plateau over time (Fig. 1 (Left)), thereby mimicking standard scaling laws (Kaplan et al., 2020; Sorscher et al., 2022). Note that for initial experimentation, we perform greedy decoding in our evaluation.

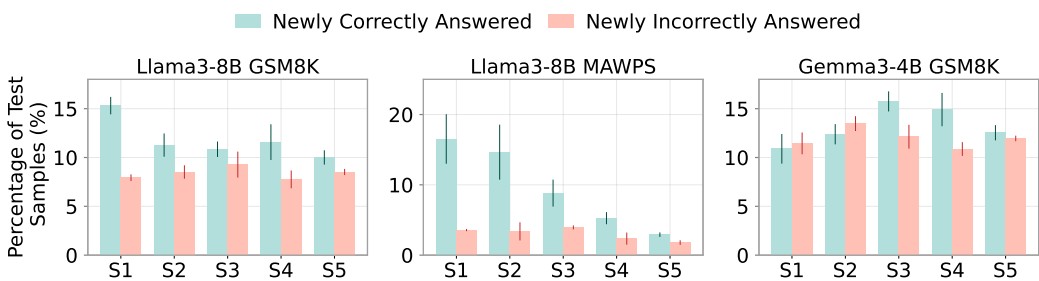

Figure 4: Even with majority voting, the addition of new training samples causes previously correctly answered samples to be incorrectly answered. `Sn` denotes Step number n when increasing the train set size.

We observe, however, that for every step at which we increase the subset-size considered, a significant number of samples that were previously correctly answered are now incorrectly answered (Fig. 1 (right column)). In fact, at every step in most settings, more than 10% of the test dataset that was previously correctly answered becomes incorrectly answered (**Newly Incorrectly Answered**). More interestingly, as we continue to increase the size of the dataset used to fine-tune the model, the number of samples that are newly correctly answered becomes very similar to the number of samples that are newly incorrectly answered, thereby resulting in marginal improvements in testing accuracy. This occurs even with RL-based training, where we perform ZeroRL of Qwen2.5-0.5B (Qwen et al., 2025) on GSM8K and Math8K (Hendrycks et al., 2021; Zeng et al., 2025) datasets using GRPO (Fig. 2). Note that we perform greedy decoding in all of these experiments during inference, such that there is no randomness during evaluation. To further demonstrate how much information is lost by increasing the size of the train data used, we compare the testing accuracy obtained by the model trained on the entire dataset (`Final Model`) against a testing metric where we assume a test sample is correctly answered if at least one of the models (trained on increasing subsets of the dataset) answers it correctly (`Union`) (see Fig. 3). We observe that the testing accuracy obtained by `Final Model` is significantly lower than that by `Union`.

### 3.1 TEST-TIME SCALING DOES NOT RESOLVE THE PROBLEM

Recent work has shown that during inference, models exhibit a high degree of non-determinism, where small changes in the sampling strategy can cause big changes in model output. Thus, it is generally recommended to sample multiple outputs using temperature sampling from the same model and perform majority voting across these outputs to get the model's prediction (Wang et al., 2023). Such sampling enhances consistency in a model's prediction, overcoming minor fluctuations that can lead to incorrect answers. Such a technique falls under a broader pool of strategies known as test-time scaling techniques, used to overcome inconsistencies and even boost performance at times.

The results in Fig. 1 are presented with naive single-sample greedy decoding. To confirm that our insights are not observed simply due to these fluctuations during inference, we repeat our evaluation with majority voting (Wang et al., 2023). Consistent with standard practice (Wang et al., 2023; Brown et al., 2024), we sample 15 outputs and perform majority voting and experiment with temperatures of 0.6 and 1 with no nucleus sampling. Since we observe worse performance when sampling with temperature 1 (sampling with high temperature has been observed to hurt performance before (Song et al., 2025)), we only present results with temperature 0.6. We observe that even with majority voting, samples continue to be incorrectly answered with the addition of more training data, and previously learned information is lost (Figs. 4 & 3).

We note that we only stick to majority voting for test-time scaling. Our sole objective is to overcome inconsistencies in model prediction, to understand if these inconsistencies and minor fluctuations are the root cause behind the phenomenon of **Newly Incorrectly Answered** samples in Fig. 1. We learned above, however, that this is not the case. There exist many test-time scaling techniques that can be used to enhance model performance, but Brown et al. (2024) show that these often provide competitive or worse performance than other test-time scaling techniques in the settings concerning math reasoning.

## 3.2 EXPERIMENTAL DETAILS

### 3.2.1 SUPERVISED FINE-TUNING

**Llama3-8b GSM8K Setting.** We fine-tune a Llama3-8B on the GSM8K dataset (Cobbe et al., 2021) using LoftQ (Li et al., 2024). Consistent with their implementation, we train for 3 epochs using a learning rate of 5e-4 with cosine decay and weight decay 0.1. Fine-tuning was performed on a machine with two Nvidia RTX4090s for 3 seeds.

**Llama3-8b MAWPS Setting.** We fine-tune a Llama3-8B on the MAWPS dataset (Koncel-Kedziorski et al., 2016) using LoftQ. We use the same hyperparameters used for the Llama3-8b-GSM8K Setting. Fine-tuning was performed on a machine with two Nvidia RTX4090s for 3 seeds.

**Gemma3-4b GSM8K Setting.** We fine-tune a Gemma3-4B on the GSM8K dataset using LoRA (Hu et al., 2022). We train for 5 epochs using a learning rate of 5e-4 with cosine decay and weight decay of 0.1. Fine-tuning was performed on a machine with two Nvidia RTX4090s for 3 seeds.

### 3.2.2 REINFORCEMENT LEARNING

**Qwen2.5-0.5B GSM8K Setting.** We fine-tune a Qwen2.5-0.5B on the GSM8K dataset in ZeroRL fashion (Zeng et al., 2025) using GRPO (Shao et al., 2024). We train for 1 epoch using a learning rate of 5e-6 with cosine decay and weight decay of 0.1. Training was performed on a machine with one NVIDIA H100 for 1 seed. We do not use any PEFT techniques.

**Qwen2.5-0.5B MATH8K Setting.** We fine-tune a Qwen2.5-0.5B on the MATH8K dataset (Hendrycks et al., 2021; Zeng et al., 2025) in ZeroRL fashion using GRPO. We train for 1 epoch using a learning rate of 5e-6 with cosine decay and weight decay of 0.1. Training was performed on a machine with one NVIDIA H100 for 1 seed. We evaluate on the MATH500 dataset (Lightman et al., 2024; Zeng et al., 2025), which is a representative subset of the MATH test set. We do not use any PEFT techniques.

## 4 LARGE LANGUAGE MODELS MAKE INCOMPLETE USE OF MATH REASONING DATA

At first glance, the results in Fig. 1 can be perceived as though additional training data is causing previously correctly answered samples to be incorrectly answered due to some seemingly conflicting information between the newly added training samples and the previously available training samples. The model is unable to disambiguate the conflict that may be present in the data. We note, however, that this is not the case. When we perform the same experiment over multiple different seeds, the newly incorrectly answered samples are very different across models. In other words, adding the *same training samples* causes different sets of test samples to be incorrectly answered. In our setting, the intersection of all the newly incorrectly answered sets by models obtained at Step 5 in Fig. 1 is less than one-fifth of all the samples that are newly incorrectly answered, as shown in Fig. 5. Due to the randomness in the newly incorrectly answered samples, it becomes evident that this occurs not because of sample-to-sample conflicts. This raises the following question: *If additional training data does not conflict with previously attained data, then why are so many previously correctly answered samples now incorrectly answered?*

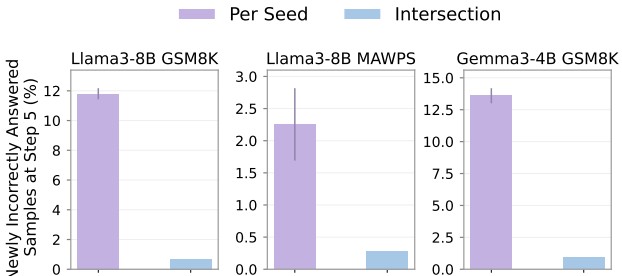

Figure 5: The intersection of the newly incorrectly answered samples at step 5 across three seeds is less than one-fifth of the average number of newly incorrectly answered samples at step 5 in all settings.

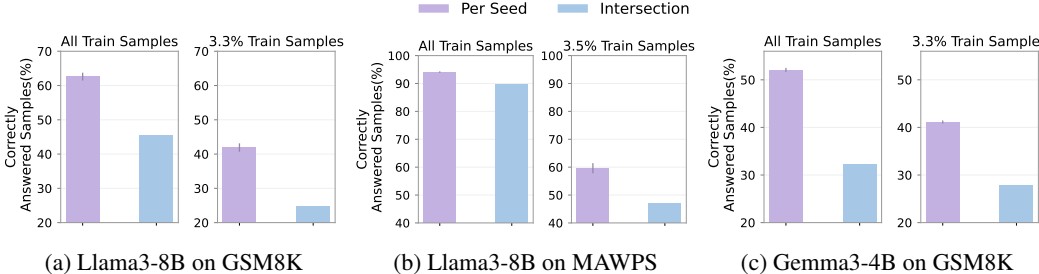

Figure 6: Fixed Set Analysis - Models trained on the same dataset but across different seeds have similar testing accuracies but correctly answer very diverse test samples.

### 4.1 FIXED SET ANALYSIS

Since additional training samples are not causing any test samples to be newly incorrectly answered due to conflicts, we hypothesize that the primary reason behind this is due to the information richness of math reasoning datasets, causing these models to learn a wide variety of different strategies and solutions. To confirm this, we train the same Llama3-8B on the entire train dataset but with different random seeds. More specifically, we train these models on the same sample set of the GSM8K dataset across three random seeds. We hypothesize and then observe that even if one were to train these models on the same fixed dataset but across *different random seeds*, they would correctly answer widely different samples within the test set, as shown in Fig. 6. In other words, the model encodes different functions over different training runs, even if it is trained on the same data. This remains true even if the sample set used to train is only a small fraction (3.3-3.5%) of the entire train set, as shown in Fig. 6. Note that we provide the intersection of correctly classified samples across the three seeds. **Thus, while the training data can be used to correctly answer many samples within the test set, each model only correctly answers a small subset of the test set. This shows that these models do not make complete use of their training data.**

### 4.2 LARGE LANGUAGE MODELS EXHIBIT HIGH PREDICTIVE MULTIPLICITY ON MATH REASONING TASKS.

In this section, we aim to better understand why these models, trained on the same data, correctly answer a very diverse set of test samples. First, we show how this phenomenon ties into existing theory and explain that these models learn different strategies per sample for math reasoning tasks. Next, we present a simple framework to show why there exist multiple models that have similar testing accuracies but correctly answer very different sets of test samples. Finally, we perform ablation studies, which show that simple changes to the training recipe can cause significant changes in the final learned function.

**Training on Math Reasoning Tasks Leads to Large Rashomon Sets.** We observe that all models trained on the same data but with different random seeds obtain almost the same testing accuracies but differ significantly on per-sample correctness, leading to a low intersection on correctly answered samples. Such predictive multiplicity has been well documented in literature, commonly referred to as the *Rashomon effect* (Breiman, 2001; Marx et al., 2020; Semenova et al., 2023; Rudin et al., 2024). The Rashomon effect is a phenomenon where multiple models obtain similar empirical risk but differ significantly in per-sample predictions. A Rashomon set (also known as $\varepsilon$-level set) is the set of all models that obtain similar empirical risk constrained by some slack $\varepsilon$.

**Definition 1 (Rashomon set)** *Let $S = \{(x_j, y_j)\}_{j=1}^n$ be a dataset and $\hat{R}_S(h)$ the empirical risk. For a baseline empirical risk minimizer $h_0 \in \arg\min_{h \in \mathcal{H}} \hat{R}_S(h)$ and $\varepsilon \geq 0$, the Rashomon set is defined as*

$$S_\varepsilon(h_0) = \{ h \in \mathcal{H} : \hat{R}_S(h) \leq \hat{R}_S(h_0) + \varepsilon \}.$$

where $\mathcal{H}$ is the hypothesis space for that task and $h$ is any model within the hypothesis space.

Note that in our setting, we define the Risk $\hat{R}$ as 0-1 error on the test dataset $S$ as follows:

$$\hat{R}_S(h) = \frac{1}{n} \sum_{i=1}^{n} \mathbf{1}[\, h(x_i) \neq y_i \,].$$

**Definition 2 (Discrepancy)** *For every model $h \in S_\varepsilon(h_0)$, we define discrepancy as*

$$\delta_\varepsilon(h) = \sum_{i=1}^{n} \mathbf{1}[\, h(x_i) \neq h_0(x_i) \,].$$

**Existence of Multiple Learnable Strategies.** Math reasoning problems are generally multi-step reasoning problems, where individual steps can be swapped or replaced entirely, while still resulting in the same correct prediction. This alone can lead to a large strategy space for a given test set. This space blows up even further when one considers the number of incorrect strategies that can be learned per sample.

We observe that across 10 different runs in the Llama3B-GSM8K setting, the average number of unique strategies per test sample was **5.32**, with the average number of unique incorrect strategies equal to **3.15**. We define strategies as the sequence of mathematical operations within the model's reasoning trace. Note that we used greedy decoding for all 10 runs.

**Definition 3 (Strategy Set)** *For every sample, we define the strategy set $K$ as the set of all unique operation sequences that yield completion (correct or incorrect) of the question.*

We extract strategies from a model's generated reasoning trace by simply extracting the operations in their appeared sequence while discarding other information. We now consider two settings that present a simple framework to show how the number of permissibly allowed models within the Rashomon set explodes with only a small increase in $|K|$.

**Setting 1 (Budget Based $\varepsilon$-Permissibility)** *We assume the simplified setting where every model $h$ within the Rashomon set makes $\delta_\varepsilon$ mistakes on baseline-correct items but does not correct baseline-incorrect items.*

Given $\hat{R}_S(h_0) = P$, the number of baseline-correct samples which can be flipped becomes $\binom{n(1-P)}{\delta_\varepsilon}$. Thus, the number of ways in which $h \in S_\varepsilon(h_0)$ can flip baseline-correct samples becomes:

$$\binom{n(1-P)}{\delta_\varepsilon} |M|^{\delta_\varepsilon}$$

for $n(1-P) \geq \delta_\varepsilon$, where $M \subset K$ is the set of incorrect strategies per sample. This gives us the number of permissible models within the Rashomon set, based on the budget $\varepsilon$.

**Setting 2 (Trades Based $\varepsilon = 0$)** *Here, we assume a setting where $\varepsilon$ tends to zero, but models disagree significantly on predictions. This is what we empirically observe in Fig. 6 (Final Model), with models attaining similar or the same testing accuracies but low intersection on correctly classified samples.*

For $\varepsilon = 0$, the necessary condition is that the number of baseline-correct flips made by any $h \in S_\varepsilon(h_0)$ must be equal to the number of baseline-incorrect flips made by that model $h$. For a given discrepancy $\delta_\varepsilon$, the number of baseline-correct and baseline-incorrect samples which can be flipped becomes $\binom{n(1-P)}{\delta_\varepsilon/2}$ and $\binom{n(P)}{\delta_\varepsilon/2}$, respectively. Thus, the number of permissible models within $S_\varepsilon(h_0)$ and equal risk $\hat{R}_S(h_0) = P$ becomes:

$$\binom{n(1-P)}{\delta_\varepsilon/2} \binom{n(P)}{\delta_\varepsilon/2} |M|^{\delta_\varepsilon/2} |K - M|^{\delta_\varepsilon/2}$$

for $n(1-P) \geq \delta_\varepsilon/2$ and $n(P) \geq \delta_\varepsilon/2$, where $M \subset K$ is the set of incorrect strategies per sample.

The GSM8K dataset has 1319 test samples, and Llama3-8B in our setting attains a Risk $\hat{R}$ of 0.38 on average. Considering these, it is evident that in both settings, minor increases in $|K|$ can cause a significant increase in $|S_\varepsilon(h_0)|$, thereby exacerbating predictive multiplicity significantly. Note that we assume that per-sample strategies are independent of each other in that a change in strategy for one sample does not impact a change in strategy for another sample.

**Ablations.** Given the large Rashomon set, it must follow that even minor changes to the training recipe while keeping the dataset and all training relevant parameters the same should result in models that differ significantly. We confirm this experimentally. Through extensive ablations in the Llama-GSM8K setting, we discover that there are two factors due to which we observe the learning of very different functions:

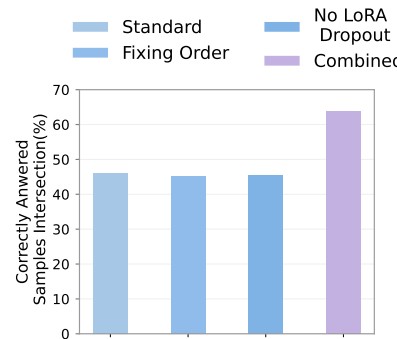

- **Sample Order.** It is well documented within the deep learning literature that the order in which samples are processed during training plays a very important role in the final function map that is eventually learned. We observe that changing this order across runs but keeping other factors constant encourages the learning of very different functions.

- **LoRA Dropout.** In our LoftQ (Li et al., 2024) setup, we maintain a LoRA dropout ratio of 0.1, whose application varies from one run to another based on the random seed. Such variance causes training to be driven in different directions, ultimately resulting in very different functions being learned.

Figure 7: Discarding or fixing factors contributing to randomness in the training process makes the model learn the same function across seeds. This is evident by the high intersection of correctly answered samples of `Combined`. `Combined` represents fixing the sample order and discarding the LoRA dropout. Intersection is taken across 3 seeds.

We observe that on fixing the order in which the model observes samples during training across three seeds, while also removing the LoRA dropout, the model learns the same function, resulting in a large intersection of correctly classified samples, as shown in Fig. 7 (the size of intersection set is equal to the size of correctly answered sample set of any of the three models). **These experiments show how small changes in the training recipe (sample order and LoRA dropout application) can cause models to learn widely different functions.**

## 5 CONCLUSION

In this work, we answer an important question: *Do large language models make complete use of the training data that they are provided?* Through comprehensive experiments across a range of models and math reasoning datasets trained using both supervised fine-tuning and reinforcement learning, we learn that they do not. First, we observe that model performance scales poorly with increasing dataset sizes, where adding more data causes smaller improvements over time due to the incorrect prediction of previously correctly answered samples. We then show that this happens not because of sample conflicts but because these models exhibit high predictive multiplicity on math reasoning tasks, where the same models trained on the same data but across different seeds learn very different functions. These models attain similar testing accuracies, but they correctly answer very different sets of test samples. Thus, while the training data can be used to correctly answer many samples within the test set, each model only correctly answers a small subset of the test set. This shows that these models do not make complete use of their training data. Finally, we tie this observation to existing theory and explain why Large Language Model training on math reasoning tasks is bound to have high predictive multiplicity.

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

## A  APPENDIX

### A.1  ROLE OF CAPACITY

To better understand the impact of capacity on incomplete use of training data, we conduct the Gemma3-4B GSM8K experiments with Gemma3-1B and Gemma3-12B. We observe that samples continue to be newly incorrectly answered with the addition of more data, despite increase in capacity (Fig. 8)

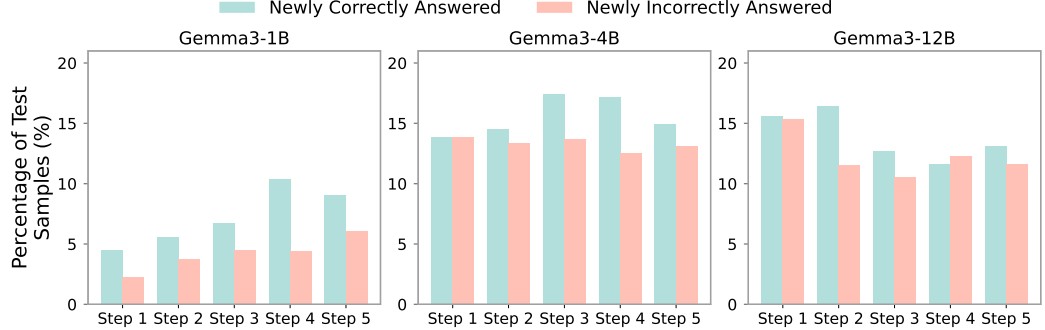

Figure 8: Samples continue to be newly incorrectly answered, despite increase in capacity

### A.2  FULL SUPERVISED FINE-TUNING

We observe that even without the use of PEFT techniques, samples are still newly incorrectly answered with the addition of more data, as shown in Fig. 9.

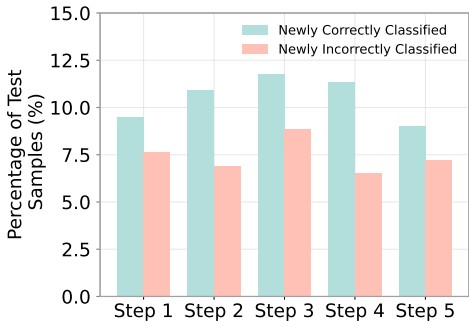

Figure 9: Samples are still newly incorrectly answered, even without the use of PEFT techniques (Llama3.2-3B on GSM8K).

