# OpenReview forum: "Large Language Models do Not Make Complete Use of Math Reasoning Data"
_ICLR.cc/2026/Conference — Submitted to ICLR 2026_

### Official Review · Reviewer_AhtP · 2025-10-21

**Soundness:** 2
**Presentation:** 3
**Contribution:** 1
**Rating:** 2
**Confidence:** 4

**Summary:**

The paper studies per-item generalization when fine-tuning LLMs on math-reasoning tasks. As training data is incrementally increased, many test items that were previously answered correctly flip to incorrect, so net accuracy improves only marginally because “newly correct” and “newly incorrect” items roughly cancel. The finding is demonstrated for SFT (e.g., Llama-3-8B, Gemma-3-4B on GSM8K/MAWPS) and RL (Qwen2.5-0.5B on GSM8K/MATH8K). The authors quantify a “Union vs Final” gap (items solved by any intermediate model vs the final model), show cross-seed “predictive multiplicity” (same data, different seeds → different test items solved), and offer a high-level explanation via a “strategy set” view of reasoning traces. Ablations suggest sample order and LoRA dropout contribute to divergence across seeds. No new training method is proposed; the work is diagnostic/observational.

**Strengths:**

– Clear empirical phenomenon: as data scales, substantial per-item flips from correct to incorrect persist, limiting net gains.

– Tracked at the item level across both SFT and RL, with multiple base models/datasets; the “Union vs Final” analysis is informative.

– Fixed-set, cross-seed analysis highlights predictive multiplicity (similar accuracy, different solved sets).

– Ablations (sample order, LoRA dropout) begin to isolate contributors to across-seed divergence.

– The problem is practically important for data scaling, curation, and evaluation of math-reasoning LMs.

**Weaknesses:**

– Limited causal explanation: the paper establishes that flips occur, but the proposed “strategy set/Rashomon” lens is descriptive and not predictive; it does not isolate necessary/sufficient causes of flipping or quantify their contributions (e.g., data conflicts vs optimization noise vs under/over-fitting vs decoding effects).

– Scope is narrow (math-reasoning, modest model sizes); it is unclear how broadly the phenomenon holds (code, multilingual, instruction-following, safety, etc.), or how it scales with much larger base models and longer training.

– No actionable solution: beyond noting order/dropout effects, the work stops short of proposing methods to reduce flips or close the Union-vs-Final gap (e.g., curriculum, data reweighting, conflict detection, checkpoint ensembling, order-invariant updates, strategy-diversity regularizers).

– Experimental controls are thin in places: small number of seeds; limited statistical testing; compute/training-length/early-stopping effects not deeply probed; decoding settings (e.g., temperature/self-consistency) only partially explored.

– The “union” signal naturally suggests simple mitigations (checkpoint ensembling across data-subset steps, EMA over training, mixture-of-checkpoints) that are not tried; without testing such baselines, the practical impact remains unclear.

– Theoretical component does not yield falsifiable predictions (e.g., when flips should increase/decrease given measurable dataset/model properties).

**Questions:**

•  What fraction of flips can be attributed to measurable data conflicts (near-duplicates with differing rationales/solutions, annotation noise) vs optimization stochasticity? Can you quantify this via conflict detection or per-sample gradient similarity analyses?

•  Can you predict which items will flip when adding data? For instance, are low-margin items (by log-prob gap), longer reasoning chains, or particular operation types more flip-prone?

•  Does the phenomenon persist with substantially larger base models and longer training to convergence? How does it scale with training steps/epochs and gradient noise scale?

•  What is the effect of decoding schemes (temperature, self-consistency, verifier-guided selection, tool-use) on flips beyond majority voting?

•  Do simple mitigations narrow the Union-vs-Final gap: (a) checkpoint ensembling across subset steps, (b) EMA of weights, (c) curriculum or order-invariant batching, (d) removing LoRA dropout and fixing order for all conditions, (e) data deduplication/cluster-balanced sampling?

•  Does full-parameter fine-tuning (no adapters) or different PEFT choices alter the flip rate?

•  Can the strategy-set view be made predictive (e.g., estimating effective strategy entropy per item) and tested against flip rates?

---

> ### Author Response · Authors · 2025-11-19
>
> We thank the reviewer for their comments. We address their concerns below:
>
> ---
>
> **W1. Limited causal explanation: the paper establishes that flips occur, but the proposed “strategy set/Rashomon” lens is descriptive and not predictive; it does not isolate necessary/sufficient causes of flipping or quantify their contributions (e.g., data conflicts vs optimization noise vs under/over-fitting vs decoding effects).**
>
>
> Predictive multiplicity in deep neural networks is a function of task underspecification. We highlight that objectives such as SFT are underspecified, thereby creating multiple models that can attain the same performance on the training objective but encode very different functions. Additionally, we address the mentioned factors below:
>
> 1) Data conflicts: Across different seeds, adding the same samples at steps causes different samples to be newly incorrectly answered (low intersection - Figure 5). This shows that sample (data) conflicts are not the reason behind the observed phenomena.
>
>
> 2) Optimization noise: As discussed in the ablations in Section 4.2, there is some optimization “noise” that is inherent to standard deep neural network training. This noise forces different learning trajectories that cause the model to learn new functions. However, this phenomenon is extremely hurtful in the settings we are concerned with and thus, important to study.
>
>
> 3) Under/over-fitting: LLM finetuning (SFT or RL) generally only happens for 1-5 epochs to avoid overfitting. However, we show that this phenomenon still occurs when one scales up the number of epochs based on the number of training samples (as requested by Reviewer z61e).
>
> | Steps| Step 1| Step 2| Step 3| Step 4| Step 5|
> | -------- | ------- | ------- | ------- | ------- | ------- |
> |Newly Correctly Answered (%)|19.71|17.51|14.94|14.33|13.42|
> |Newly Incorrectly Answered (%)|9.93|11.9|13.42|12.05|13.72|
>
>
> Here, we trained the model on the smallest subset with 1 epoch and increased the number of epochs by 1 each time we add more data.
>
> We observe the same behavior. We will be happy to run additional experiments on request by the reviewer.
>
>
> 4) Decoding effects: In our first analysis, we perform greedy decoding to eliminate any non-determinism in the inference process. However, even with temperature sampling and utilization of test-time scaling approaches like majority voting, we still observe the same problem (Figure 4).
>
> ---
>
> **W2. Scope is narrow (math-reasoning, modest model sizes); it is unclear how broadly the phenomenon holds (code, multilingual, instruction-following, safety, etc.), or how it scales with much larger base models and longer training.**
>
>
> We would like to clearly state that the scope of this work is mathematical reasoning as defined in the title. We have shown results with larger base models and longer training in other responses to your queries.
>
> ---
>
> **W3. No actionable solution: beyond noting order/dropout effects, the work stops short of proposing methods to reduce flips or close the Union-vs-Final gap (e.g., curriculum, data reweighting, conflict detection, checkpoint ensembling, order-invariant updates, strategy-diversity regularizers).**
>
> We thank the reviewer for their comment. We believe this work is the first step in a sequence of works that help large language models make better use of training reasoning data. This work identifies the problem through extensive empirical evidence. We strongly believe an effective and usable solution to this problem is challenging enough that it would warrant being its own paper.
>
> ---
>
> **W4. Experimental controls are thin in places: small number of seeds; limited statistical testing; compute/training-length/early-stopping effects not deeply probed; decoding settings (e.g., temperature/self-consistency) only partially explored.**
>
> We disagree with the reviewer on the number of seeds. Our analysis is for 3 random seeds, which is standard in deep learning and other works in this domain.
>
> For compute variations, we scale up the number of epochs in our GSM8K result above. We also note that in these settings, early-stopping effects are not generally studied. This is because the total number of training epochs is between 1-5. However, if the reviewer feels strongly about this, we can rerun our experiments with early stopping based on any references the reviewer can provide.
>
> We also show that our findings are consistent with greedy and temperature sampling-based decoding, along with test-time scaling. Thus, we must disagree with the reviewer on this point.

---

> > ### Author Response · Authors · 2025-11-19
> >
> > **W5. The “union” signal naturally suggests simple mitigations (checkpoint ensembling across data-subset steps, EMA over training, mixture-of-checkpoints) that are not tried; without testing such baselines, the practical impact remains unclear.**
> >
> > We thank the reviewer for this suggestion. We did consider two methods that would consider outputs from all models:
> >
> > *Model-based majority voting*: Instead of sampling multiple outputs from the same model, sample greedy outputs from intermediate models and then perform majority voting.
> >
> > *Confidence-based* voting: Pick the most confident output from all intermediate models.
> >
> > Additionally, we repeated these methods but with multiple models trained on full datasets with different seeds (instead of intermediate models with the same seed).
> >
> > We note that none of these methods overcame the issue of samples being newly incorrectly answered (and did not notably reduce the number of newly incorrectly answered samples.)  We will be happy to include these unsuccessful attempts in the appendix if the reviewer believes it will help inform readers of what does not work and thus should not be attempted in the same way.
> >
> > ---
> >
> > **W6. Theoretical component does not yield falsifiable predictions (e.g., when flips should increase/decrease given measurable dataset/model properties).**
> >
> > We disagree with the reviewer on this point. The number of permissible models will reduce when the number of strategies learned per sample drops.
> >
> > The underlying problem the theory highlights is that models that are similarly trained learn very different functions. This implies that standard training objectives are underspecified. This observation can aid in the creation of better training objectives, especially for reasoning tasks.
> >
> > ---
> >
> > **Q1. What fraction of flips can be attributed to measurable data conflicts (near-duplicates with differing rationales/solutions, annotation noise) vs optimization stochasticity? Can you quantify this via conflict detection or per-sample gradient similarity analyses?**
> >
> > Yes we can. The fraction of flips attributed to measurable data conflicts is almost zero. We know this because the intersection of newly incorrectly classified samples at different steps across seeds is close to zero (Figure 5 in the originally submitted manuscript).
> >
> > Almost all flips occur due to optimization stochasticity and in our ablation analysis, we show that making the process deterministic by eliminating factors contributing to optimization stochasticity increases the intersection of correctly answered samples perfectly (Figure 7 in the original manuscript).
> >
> > ---
> >
> > **Q2. Can you predict which items will flip when adding data? For instance, are low-margin items (by log-prob gap), longer reasoning chains, or particular operation types more flip-prone?**
> >
> >
> > We thank the reviewer for their comment.
> >
> > In our Llama3-8B GSM8K setup, we observe that about 603 out of 1319 test samples are newly incorrectly answered at least once across all steps. We observe that the average generated token probability, margin, and solution length for these samples as follows:
> >
> >
> > Token probability: 0.9355, Margin: 0.891, solution length: 109.7 tokens
> >
> >
> > For the remaining 716 samples, 221 of them are never correctly answered so we discard them from our analysis. For the remaining samples:
> >
> > Token probability: 0.945, Margin: 0.906, solution length: 91.54 tokens
> >
> > We observe that while there isn’t a significant difference in token probability/margin, questions with longer generated solutions are more likely to be flipped to “newly incorrectly answered”.
> >
> > We will be happy to include these results in the revised version of the manuscript based on the reviewers’ request.
> >
> > ---
> >
> > **Q3. Does the phenomenon persist with substantially larger base models and longer training to convergence? How does it scale with training steps/epochs and gradient noise scale?**
> >
> >
> > To understand the impact of scale, we repeat the Gemma3-4B GSM8K Experiment with Gemma3-12B instead. We observe the same phenomenon as shown below:
> >
> > Gemma3-4B (Test Accuracy when trained on all data: 52.46%):
> >
> > | Steps| Step 1| Step 2| Step 3| Step 4| Step 5|
> > | -------- | ------- | ------- | ------- | ------- | ------- |
> > |Newly Correctly Answered (%)|13.87|14.48|17.44|17.21|14.94|
> > |Newly Incorrectly Answered (%)|13.87|13.34|13.72|12.51|13.12|
> >
> >
> > Gemma3-12B (Test Accuracy when trained on all data: 72.93%):
> >
> > | Steps| Step 1| Step 2| Step 3| Step 4| Step 5|
> > | -------- | ------- | ------- | ------- | ------- | ------- |
> > |Newly Correctly Answered (%)|15.62|16.45|12.66|11.6|13.12|
> > |Newly Incorrectly Answered (%)|15.31|11.52|10.54|12.28|11.6|
> >
> > We have included this result in the appendix of the revised version of our manuscript (Figure 9).
> >
> > In a previous response to your questions, we have shown that the phenomenon still persists with longer training with more data.

---

> > > ### Author Response · Authors · 2025-11-19
> > >
> > > **Q4. What is the effect of decoding schemes (temperature, self-consistency, verifier-guided selection, tool-use) on flips beyond majority voting?**
> > >
> > > We note that we already used temperature sampling for test time scaling. It is important to note that for just a single sample, temperature sampling will yield even more inconsistent outputs. Hence, it does not make much sense to do temperature sampling, unless paired with test-time scaling as we did.
> > >
> > > We also note that generally, self-consistency and majority voting are synonymous in deep learning literature (Brown et. al. 2024). They both pick the most common final answer.
> > >
> > > Finally, we note that for the tasks we are concerned with, majority voting has been shown to be the most optimal (Brown et. al. 2024) compared to other test time scaling techniques. Additionally, self-consistency/majority voting would be the best at overcoming randomness in the generation process. We also note that it is infeasible to perform experiments on all decoding schemes.
> > >
> > >
> > > Brown et. al. 2024 Large Language Monkeys: Scaling Inference Compute with Repeated Sampling
> > >
> > > ---
> > >
> > > **Q5. Do simple mitigations narrow the Union-vs-Final gap: (a) checkpoint ensembling across subset steps, (b) EMA of weights, (c) curriculum or order-invariant batching, (d) removing LoRA dropout and fixing order for all conditions, (e) data deduplication/cluster-balanced sampling?**
> > >
> > > We thank the reviewer for this suggestion. We did consider two methods that would consider outputs from all models:
> > >
> > > *Model-based majority voting*: Instead of sampling multiple outputs from the same model, sample greedy outputs from intermediate models and then perform majority voting.
> > > *Confidence-based* voting: Pick the most confident output from all intermediate models.
> > >
> > > Additionally, we repeated these methods but with multiple models trained on full datasets with different seeds (instead of intermediate models with the same seed).
> > >
> > > We note that none of these methods overcame the issue of samples being newly incorrectly answered (and did not notably reduce the number of newly incorrectly answered samples.)  We will be happy to include these unsuccessful attempts in the appendix if the reviewer believes it will help inform readers of what does not work and thus should not be attempted in the same way.
> > >
> > > Additionally, we note that removing LoRA dropout and fixing order would only work for overcoming fixed set variability, as shown in section 4.2. With the addition of more data batches, this would no longer work due to additional training creating new variations in the learned function.
> > >
> > > ---
> > >
> > > **Q6. Does full-parameter fine-tuning (no adapters) or different PEFT choices alter the flip rate?**
> > >
> > > We note that our RL experiments for MATH8k and GSM8K with Qwen2.5-0.5B do not make use of PEFT techniques. However, we still observe newly incorrectly answered samples across steps (flips).
> > >
> > > We also run full supervised fine-tuning (No PEFT/adapters) on Llama3.2-3B on GSM8K for 1 seed and observe a consistent phenomenon:
> > >
> > > | Steps| Step 1| Step 2| Step 3| Step 4| Step 5|
> > > | -------- | ------- | ------- | ------- | ------- | ------- |
> > > |Newly Correctly Answered (%)|9.48|10.92|11.75|11.37|9.02|
> > > |Newly Incorrectly Answered (%)|7.66|6.9|8.87|6.52|7.2|
> > >
> > > We have included this result in the appendix of the revised version of our manuscript (Figure 8).
> > >
> > > We also note that we use two PEFT techniques in our work: LoftQ and LoRA.
> > >
> > > Our observations are consistent across all setups.
> > >
> > > ---
> > >
> > > **Q7. Can the strategy-set view be made predictive (e.g., estimating effective strategy entropy per item) and tested against flip rates?**
> > >
> > > We are unsure if we have correctly understood the reviewer’s question. From our understanding, it is evident that the greater the number of learnable strategies per sample/item, the higher the chance that a sample can flip from correctly answered to incorrectly answered and vice versa. We hope this helps.

---

> ### Author Response · Authors · 2025-11-24
>
> Dear Reviewer AhtP,
>
> Kindly let us know if our rebuttal has addressed your concerns or if you have any additional concerns regarding our work.
>
> We thank you for your time and consideration,
>
> Authors

---

> > ### Comment · Reviewer_AhtP · 2025-11-25
> >
> > I appreciate the detailed responses, additional experiments, and the clarification of several points. However, I still have two remaining concerns:
> >
> > 1. **Causal explanation and predictivity remain largely qualitative.**
> >    The rebuttal helpfully clarifies the roles of data conflicts, optimization noise, under/over-fitting, and decoding, and adds new analyses (for example, longer training and item-level statistics such as margin and solution length). That said, the overall explanation still feels largely qualitative and post hoc:
> >    * The conclusion that data conflicts play essentially no role is still inferred indirectly from low cross-seed overlap of newly-incorrect items, rather than from direct conflict detection (near-duplicates, label inconsistencies) or gradient-similarity analysis. Low intersection alone does not rigorously rule out structured conflicts in specific regions of the data.
> >    * The “strategy set / Rashomon” view remains descriptive: while you now report some item-level statistics (for example, solution length) and note that longer solutions are more flip-prone, there is still no quantitative test of the central theoretical claim (for example, estimating per-item strategy entropy and relating it to flip rates), nor a model that predicts flip probability from measurable features.
> >    * More broadly, there is still no quantitative causal decomposition (for example, what fraction of flips is attributable to optimization noise vs under/over-fitting vs decoding) and no clearly falsifiable prediction derived from the theory that is empirically tested.
> >
> >    Taken together, the theory and experiments provide a convincing qualitative narrative but do not yet rise to a measured, predictive causal account of when and why flips occur.
> >
> > 2. **Mitigations and practical impact remain limited despite useful negative results.**
> >    It is very helpful that you now report having tried several natural “union-style” mitigations, such as majority voting and confidence-based selection across intermediate checkpoints and across seeds, and that these methods do **not** substantially reduce the number of newly-incorrect items. This negative result strengthens the diagnostic message, but it also highlights that:
> >    * The most immediate mitigations suggested by the union signal appear ineffective in your experiments, and there is still no training or inference modification that demonstrably narrows the Union-vs-Final gap or lowers flip rates on the main benchmarks.
> >    * The promising ablation that fixes sample order and removes LoRA dropout is analyzed primarily in terms of cross-seed intersection and “fixed-set variability”, but is not developed into an end-to-end “stabilized training” baseline with quantitative impact on the core metrics (Newly-Incorrect counts, Union-vs-Final).
> >    * Other natural interventions motivated by your analysis, such as weight-space EMA, parameter-space checkpoint ensembling, curriculum or order-invariant batching, or more principled de-duplication/reweighting schemes, are discussed only at a high level (if at all) and are not empirically evaluated.
> >
> >    As a result, the work remains, in my view, primarily diagnostic: it convincingly documents a phenomenon and reports useful negative evidence about simple ensembling and voting approaches, but stops short of identifying even a simple mitigation that is shown to help in practice. I believe the paper would be significantly strengthened either by developing at least one such mitigation into a quantitative baseline, or by more explicitly framing the current contribution as an observational study of per-item generalization and predictive multiplicity.

---

> ### Author Response · Authors · 2025-11-26
>
> Thank you for your follow-up concerns. We address them below:
>
> **Taken together, the theory and experiments provide a convincing qualitative narrative but do not yet rise to a measured, predictive causal account of when and why flips occur.**
>
> We note that predictive multiplicity in practice is known to be caused due to:
>
> 1. Underspecification in the training objective: Different functions can attain similar risk/loss on the training objective - this is exactly what we observed and mentioned in our previous response.
> 2. A rich data distribution - long sequences, diverse reasoning chains and large vocabulary.
>
>
> Thus, small changes in factors like data order can cause training to encourage these models to learn very different sets of functions.
> Additionally, in practice, one does not fine-tune large language models for many epochs to avoid overfitting.
>
> Finally, kindly note that while the samples that are newly incorrectly answered do have longer reasoning chains, they also have more operations, allowing for more usable and learnable strategies for that sample and increased possibility for flips:
>
> 603 “newly incorrectly answered”: 3.35 operations on average.
>
> 495 never “newly incorrectly answered”: 2.97 operations on average.
>
> Kindly also note that with the addition of the same data points at the same step across different seeds, conflicts due to near duplicates and label noise should cause the same samples to be newly incorrectly answered for all or at least most seeds but we do not observe this.
>
> ---
>
> **It convincingly documents a phenomenon and reports useful negative evidence about simple ensembling and voting approaches, but stops short of identifying even a simple mitigation that is shown to help in practice. I believe the paper would be significantly strengthened either by developing at least one such mitigation into a quantitative baseline, or by more explicitly framing the current contribution as an observational study of per-item generalization and predictive multiplicity.**
>
> Thank you for your comment. Our current frame already notes the contribution as an insightful study of per-item generalization and predictive multiplicity. The paper, as it currently stands, highlights very important novel negative aspects of scaling in large language models that are useful to know for the research community.

---

### Official Review · Reviewer_NUTt · 2025-10-28

**Soundness:** 4
**Presentation:** 3
**Contribution:** 3
**Rating:** 6
**Confidence:** 4

**Summary:**

The paper foucs on one question do LLMs make complete use of the training data and the short answer is no which they show on different smaller models using and number of math datasets from GSM8k to Hendryks math. The models are mostly small with Lamma3 8B or Gemma 4b.

**Strengths:**

The study has interesting findings and opens a research agenda for smaller models. I find it interesting to know.  Nice focus on one and important big questions.

**Weaknesses:**

I wonder more about the reason, is it the size and limited parameter so that the model has to make trade-offs.

The paper would be strong if the authors could show how this scales between models, you do not need to go to bigger models but could also smaller if there is still a signal found.

**Questions:**

The question seems to be difficult: is it related to size? You use 8b and 4b models, do you observe differences? They have just different capacities and I guess the smaller models have to cramp the knowledge into a few parameters? Does this hypothesis hold or is it something else? They are a bit from different generation of models.

---

> ### Author Response · Authors · 2025-11-19
>
> We thank the reviewers for their comments and for sharing their interest in our work and the question we aim to answer. We address their concerns below:
>
> ---
>
> **The question seems to be difficult: is it related to size? You use 8b and 4b models, do you observe differences? They have just different capacities and I guess the smaller models have to cramp the knowledge into a few parameters? Does this hypothesis hold or is it something else? They are a bit from different generation of models.**
>
> We thank the reviewer for this question. To understand the role of capacity, we repeat the Gemma3-4B GSM8K experiments with Gemma3-1B and Gemma3-12B. This keeps the model family and model generation the same, but only varies the capacity. However, we still observe the same phenomenon as shown below:
>
> Gemma3-1B (Test Accuracy when trained on all data: 18.57%):
>
> | Steps| Step 1| Step 2| Step 3| Step 4| Step 5|
> | -------- | ------- | ------- | ------- | ------- | ------- |
> |Newly Correctly Answered (%)|4.47|5.53|6.75|10.39|9.02|
> |Newly Incorrectly Answered (%)|2.27|3.71|4.47|4.4|6.07|
>
>
> Gemma3-4B (Test Accuracy when trained on all data: 52.46%):
>
> | Steps| Step 1| Step 2| Step 3| Step 4| Step 5|
> | -------- | ------- | ------- | ------- | ------- | ------- |
> |Newly Correctly Answered (%)|13.87|14.48|17.44|17.21|14.94|
> |Newly Incorrectly Answered (%)|13.87|13.34|13.72|12.51|13.12|
>
>
> Gemma3-12B (Test Accuracy when trained on all data: 72.93%):
>
> | Steps| Step 1| Step 2| Step 3| Step 4| Step 5|
> | -------- | ------- | ------- | ------- | ------- | ------- |
> |Newly Correctly Answered (%)|15.62|16.45|12.66|11.6|13.12|
> |Newly Incorrectly Answered (%)|15.31|11.52|10.54|12.28|11.6|
>
>
> We note that the smallest model (1B) obtains fewer newly incorrectly answered (%) because it does not answer most test samples correctly in the first place. Interestingly, at high capacity (12B), we still observe a significant number of samples being newly incorrectly answered.
>
> We have included this result in the appendix of the revised version of our manuscript (Figure 8).
>
> We strongly believe that the reason for samples being “newly incorrectly answered” is underspecification in the learning objective, where many models can be permissibly good and similar during training on an objective like SFT (which simply optimizes for next token loss). While these models may have similar performances on the training objective, the model still doesn’t learn to focus on more specific goals that are critical for generalizability and stability like operation sequence, word-operation mapping, etc. As a result, many different functions can be learned.
>
> If one were to teach these models to learn more specific goals relevant to reasoning, it is likely the models would learn much more similar functions across steps and seeds.
>
> ---
>
> We will be happy to address any additional concerns the reviewer may have with the paper or our response. We thank them for their time and for sharing their interest in our work.

---

> ### Author Response · Authors · 2025-11-24
>
> Dear Reviewer NUTt,
>
> Kindly let us know if our rebuttal has addressed your concerns or if you have any additional concerns regarding our work.
>
> We thank you for your time and consideration,
>
> Authors

---

### Official Review · Reviewer_z61e · 2025-10-31

**Soundness:** 3
**Presentation:** 3
**Contribution:** 4
**Rating:** 6
**Confidence:** 4

**Summary:**

This paper investigates how large language models (LLMs) improve performance during fine-tuning on mathematical reasoning tasks. Contrary to the assumption that more data monotonically improves performance, the authors find that adding additional training samples often causes models to forget previously capabilities — around 10–15% of correctly answered test samples become incorrect after adding more data. This behavior persists under both supervised fine-tuning (SFT) and reinforcement learning (RLVR) training paradigms, and even with test-time scaling techniques such as majority voting. The authors attribute this phenomenon to high predictive multiplicity (the Rashomon effect), where models trained on the same data but with different random seeds learn significantly different functions, leading to diverse sets of correctly answered questions. The paper argues that LLMs make incomplete use of math reasoning data because of this multiplicity and sensitivity to training randomness.

**Strengths:**

1. Interesting empirical finding: The paper opens the black-box performance in standard scaling-law assumptions and highlights a new failure mode of data scaling, which is crucial to understanding generalization in LLM finetuning.

2. Comprehensive experimental design: The study spans multiple models (Llama3-8B, Gemma3-4B, Qwen2.5-0.5B) and training paradigms (SFT and RL), making the findings robust across architectures and methods.

3. Clear visualization and metrics: Figures and fixed-set analyses (e.g., intersection analysis across random seeds) effectively communicate the instability in model predictions.

**Weaknesses:**

1. The claim "Large language models do not make complete use of training data" is inaccurate to summary the paper conclusion and the term "make use" is ambiguous. The main gist of the paper is generalization capacity rather than training data/sample efficiency, and large language model can certainly "make use" of training data by memorizing them correctly.

2. The message from the theoretical calculation of mode density is unclear. While it is unclear why a "strategy set" concept is necessary to introduce into the calculation, there is also a lack of explanation on the result of the calculation and how it implies or adds to the empirical findings.

3. The paper does not distinguish the proposed forgetting phenomenon from the traditional overfitting phenomenon, as the paper neglected another important factor of training epochs (in each "step") that also affect model performances. While for different "steps" the model is trained on different training sets, the model also goes through different epochs on training data and the forgetting phenomenon may be attributed to severe overfitting on small datasets.

Can you provide more details on how many epochs the model went through in each training stage, and how does model performance / newly correct/incorrect number changes in the process? Can you confirm that for every training set respectively, the training epoch is tuned optimally?

4. There is shortage of fine-grained analysis on the training / newly-correct / newly-incorrect problems regarding topics, difficulty levels, solution length, etc.

**Questions:**

1. Can you elaborate on the weaknesses above?
2. What does "random seed" in the paper control in the training process and how are the data shuffled? This term is blurry in the original paper.

---

> ### Author Response · Authors · 2025-11-19
>
> We thank the reviewer for taking the time to present detailed comments. We are grateful that they recognize the key contributions of our paper. We address their concerns below:
>
> ---
>
> **W1. The claim "Large language models do not make complete use of training data" is inaccurate to summary the paper conclusion and the term "make use" is ambiguous. The main gist of the paper is generalization capacity rather than training data/sample efficiency, and large language model can certainly "make use" of training data by memorizing them correctly.**
>
>
> We apologize for this ambiguity. We will work to resolve this ambiguity in the text:
>
> Our claim about incomplete use is based on the evidence:
>
> a) Increasing the dataset size causes these models to incorrectly answer previously correctly answered samples, implying that previous train samples that helped to answer some test samples are no longer helping or being **used** properly. This is what we mean by not “making use”. This is evidenced by the high “Union” accuracy in Figure 3 and significantly lower “Final Model” accuracy in Figure 3.
>
> b) Training on the **same dataset** across different seeds results in models that learn different functions that can only answer a part of the test set correctly, implying that these models cannot make complete use of the train set in one run (Figure 6).
>
> We agree with the reviewer on the following point - it is not about sample efficiency - each data point is very rich and we are not claiming that we need more data for better performance. We simply stress that these models cannot make good use of this rich data completely.
>
> We are happy to hear any alternative wording the reviewer may suggest or any additional points the reviewer wants to discuss. We thank them for their time.
>
> ---
>
> **W2. The message from the theoretical calculation of mode density is unclear. While it is unclear why a "strategy set" concept is necessary to introduce into the calculation, there is also a lack of explanation on the result of the calculation and how it implies or adds to the empirical findings.**
>
> Our theoretical analysis presents a simple analytical tool to understand how quickly the Rashomon set can explode with small increases in the number of strategies the model can follow. It is important to consider a “strategy set” as this can blow up the number of permissible models in the Rashomon set. The root cause behind high predictive multiplicity is that the total number of learnable strategies per sample is large (correct or incorrect). We discussed this from lines 432-436 in the original text.
>
> It is important to show that Rashomon Sets are large because it highlights how underspecified modern training objectives are. These models are capable of learning/understanding a lot more, as is shown in our experimental results, but struggle to do so. This insight can aid researchers in proposing new solutions in the future.
>
> Note that we are happy to expand on this if the reviewer believes it will help the readers.
>
> ---
>
> **W3. The paper does not distinguish the proposed forgetting phenomenon from the traditional overfitting phenomenon, as the paper neglected another important factor of training epochs (in each "step") that also affect model performances. While for different "steps" the model is trained on different training sets, the model also goes through different epochs on training data and the forgetting phenomenon may be attributed to severe overfitting on small datasets.**
>
> We thank the reviewer for this comment. We repeat the same experiment but scale the number of epochs based on the number of total training samples. Below, we trained the model on the smallest subset with 1 epoch and increased the number of epochs by 1 each time we add more data:
>
> | Steps| Step 1| Step 2| Step 3| Step 4| Step 5|
> | -------- | ------- | ------- | ------- | ------- | ------- |
> |Newly Correctly Answered (%)|19.71|17.51|14.94|14.33|13.42|
> |Newly Incorrectly Answered (%)|9.93|11.9|13.42|12.05|13.72|
>
> We observe the same behavior.
>
> Kindly note that when fine-tuning LLMs on such tasks, these models are generally trained for 1-5 epochs, and thus, to avoid overfitting, we did not want to scale the number of epochs by too much. Originally, in our Llama3-8B GSM8K setup, we train all intermediate models for 3 epochs (lines 274-277 in the original script).

---

> > ### Author Response · Authors · 2025-11-19
> >
> > **W4. There is shortage of fine-grained analysis on the training / newly-correct / newly-incorrect problems regarding topics, difficulty levels, solution length, etc.**
> >
> > We agree with the reviewer on this point and thank them for proposing this idea. However, we note that the intersection of correctly/incorrectly answered samples at steps **across seeds** is very low (Figure 5). This happens despite adding the **same samples** at each step across seeds. This tells us that there are no discernible patterns regarding topics/samples that are newly correctly/incorrectly classified with the addition of specific data. We do, however, discuss difficulty level and solution length of newly incorrectly answered samples in contrast to other samples below, according to the reviewer’s comment:
> >
> > First, we identify that 603/1319 test samples in the Llama3-8B GSM8K are “newly incorrectly answered” at some step. This accounts for 45.7% of the test set.
> >
> > \
> > For these 603 samples:
> >
> > Average solution length: 109.7 tokens.
> >
> > Average token probability: 0.935.
> >
> > Average token margin: 0.891
> >
> > \
> > Among the remaining 716 samples that are never “newly incorrectly answered”, 221 of them are never correctly answered so we discard them from our analysis. For the remaining samples:
> >
> > \
> > Average solution length: 91.54 tokens.
> >
> > Average token probability: 0.945.
> >
> > Average token margin: 0.906.
> >
> > \
> > The latter subset has fewer tokens and is slightly easier based on model output confidence and margin, but next, we show the same statistics for the 100 samples that are most frequently newly incorrectly answered (across steps):
> >
> > \
> > Average solution length: 118.7 tokens.
> >
> > Average token probability: 0.931.
> >
> > Average token margin: 0.884.
> >
> > \
> > We also compute ground truth token probabilities and ground truth solution lengths, and we found that for the 603 “newly incorrectly answered” subsets:
> >
> > \
> > Avg. token length: 112.32 tokens.
> >
> > Avg. token probability: 0.60.
> >
> > \
> > While for the 495 samples that are never “newly incorrectly answered” (but answered correctly at some step):
> >
> > \
> > Avg. token length: 92.21 tokens.
> >
> > Avg. token probability: 0.649.
> >
> > \
> > From these experiments, it is evident that “newly incorrectly answered” samples are harder on average (based on solution length and token probability/confidence/margin). Based on the reviewers’ comments, we will be happy to include these results in the revised version of our paper.
> >
> > ---
> >
> >
> > **Q2. What does "random seed" in the paper control in the training process and how are the data shuffled? This term is blurry in the original paper.**
> >
> > We apologize for not clarifying this. Random seed controls all processes that exhibit randomness within our training script. In our Llama3-8B GSM8K script, this is the LoRA dropout and the sample order (data shuffling) as mentioned in Section 4.2 (Ablations). One can choose to fix the random seed and only vary the sample order (data seed) through the huggingface trainer config (we do this in our ablations). Please note that we will open-source our code upon acceptance so it will be easy for practitioners to run these ablations themselves.
> >
> > ---
> >
> > We will be happy to address any additional concerns the reviewer may have.

---

> ### Author Response · Authors · 2025-11-24
>
> Dear Reviewer z61e,
>
> Kindly let us know if our rebuttal has addressed your concerns or if you have any additional concerns regarding our work.
>
> We thank you for your time and consideration,
>
> Authors

---

### Official Review · Reviewer_B3Vw · 2025-11-01

**Soundness:** 2
**Presentation:** 3
**Contribution:** 2
**Rating:** 2
**Confidence:** 4

**Summary:**

This paper investigates the phenomenon where fine-tuning LLMs on increasing amounts of mathematical reasoning data does not lead to monotonic improvements on a fixed test set. The authors demonstrate that as more training data is added, a significant portion (10-15%) of previously correctly answered test samples become incorrectly answered. This "churn" mitigates the net performance gain. The authors attribute this behavior to high "predictive multiplicity", where models trained on the same data with different random seeds learn vastly different functions. These models achieve similar overall test accuracies but agree on a surprisingly small subset of correctly answered questions. The paper supports these claims with experiments using both SFT and RL on models like Llama3, Gemma3, and Qwen2.5 across datasets such as GSM8K, MAWPS, and MATH.

**Strengths:**

1.  The paper compellingly demonstrates a non-trivial and counter-intuitive aspect of scaling laws at the sample level.

2. The experiments are systematically conducted across multiple models, datasets, and training paradigms (SFT and RL). The use of multiple seeds is crucial and well-executed.

3. The authors proactively address potential confounding factors by showing the phenomenon persists even with majority voting, a standard technique to reduce prediction variance.

**Weaknesses:**

1. The paper frames its findings as LLMs "not making complete use of data." This is a very strong claim. An alternative, and perhaps more standard, interpretation is that this is a natural consequence of stochastic gradient-based optimization in a high-dimensional, non-convex landscape. The optimizer is finding a new local minimum to accommodate new data, which inevitably shifts the decision boundary for old data. The paper does not sufficiently differentiate its findings from the well-established field of catastrophic forgetting or continual learning.

2.  The analysis in Section 4.2, which introduces the Rashomon set, feels like a post-hoc application of a known concept rather than a deep analysis. The assumptions made are overly simplistic and the calculations for the "number of permissible models" are combinatorial thought experiments. They do not provide a rigorous link between the training dynamics (e.g., optimizer, learning rate) and the size of the observed Rashomon set.

3.  The experiments primarily use PEFT methods. It is unclear if this phenomenon is exacerbated by the constrained updates of PEFT. A comparison with full fine-tuning would be necessary to claim this is a general property of LLM training.

**Questions:**

1.  How do the authors distinguish their findings from the classic problem of "catastrophic forgetting" in continual learning? While the setting isn't strictly continual, adding batches of data and observing performance degradation on previously learned samples seems highly related.

2.  The "Union" accuracy in Figure 3 is significantly higher than the final model's accuracy. This strongly suggests that ensembling the models from each training step would be a very effective mitigation strategy. Did the authors consider exploring this or any other method to address the information loss?

3.  Could the authors comment on the role of PEFT in their observations? Is it possible that the low-rank updates inherent to methods like LoRA make the model more susceptible to this "churn," as the capacity to integrate new information without disrupting old knowledge is constrained? How would these results compare to full fine-tuning?

4. The theoretical analysis in Section 4.2 makes a strong simplifying assumption in "Setting 1" that models in the Rashomon set *only* make mistakes on baseline-correct items and never correct baseline-incorrect ones. This seems unrealistic. How does the analysis hold if this assumption is relaxed to a more plausible trade-off?

---

> ### Author Response · Authors · 2025-11-19
>
> We thank the reviewer for their detailed comments and for taking the time to improve our work. We address their concerns below:
>
> ---
>
> **Q1. How do the authors distinguish their findings from the classic problem of "catastrophic forgetting" in continual learning? While the setting isn't strictly continual, adding batches of data and observing performance degradation on previously learned samples seems highly related.**
>
> We thank the reviewer for this question.
>
> We clearly note that our observations are different from the catastrophic forgetting phenomena in continual learning literature. This is because models trained on the **same dataset** (without adding new batches of data) still learn very different functions (Section 4.1). We will ensure to clarify this in related works.
>
> ---
>
> **Q2. The "Union" accuracy in Figure 3 is significantly higher than the final model's accuracy. This strongly suggests that ensembling the models from each training step would be a very effective mitigation strategy. Did the authors consider exploring this or any other method to address the information loss?**
>
> We thank the reviewer for this suggestion. We did consider two methods that would consider outputs from all models:
>
> *Model-based majority voting*: Instead of sampling multiple outputs from the same model, sample greedy outputs from intermediate models and then perform majority voting.
>
> *Confidence-based* voting: Pick the most confident output from all intermediate models.
>
> Additionally, we repeated these methods but with multiple models trained on full datasets with different seeds (instead of intermediate models with the same seed).
>
> We note that none of these methods overcame the issue of samples being newly incorrectly answered (and did not notably reduce the number of newly incorrectly answered samples.)  We will be happy to include these unsuccessful attempts in the appendix if the reviewer believes it will help inform readers of what does not work and thus should not be attempted in the same way.
>
> ---
>
> **Q3. Could the authors comment on the role of PEFT in their observations? Is it possible that the low-rank updates inherent to methods like LoRA make the model more susceptible to this "churn," as the capacity to integrate new information without disrupting old knowledge is constrained? How would these results compare to full fine-tuning?**
>
> We note that not all our experimental results make use of PEFT techniques. Our results on Math8K and GSM8K with RL do not make use of PEFT techniques. Considering that both PEFT and non-PEFT training techniques exhibit the same behavior leads us to conclude that PEFT alone is not the root cause behind our observations. Nevertheless, we run a Llama3.2-3b GSM8K experiment with full supervised fine-tuning on one seed to show that this behavior is consistent.
>
> | Steps| Step 1| Step 2| Step 3| Step 4| Step 5|
> | -------- | ------- | ------- | ------- | ------- | ------- |
> |Newly Correctly Answered (%)|9.48|10.92|11.75|11.37|9.02|
> |Newly Incorrectly Answered (%)|7.66|6.9|8.87|6.52|7.2|
>
> We have included this result in the appendix of the revised version of our manuscript (Figure 9).
>
> ---
>
> **Q4. The theoretical analysis in Section 4.2 makes a strong simplifying assumption in "Setting 1" that models in the Rashomon set only make mistakes on baseline-correct items and never correct baseline-incorrect ones. This seems unrealistic. How does the analysis hold if this assumption is relaxed to a more plausible trade-off?**
>
>
>
> Through Setting 2 (present in the original version), we offer a setting where trades occur.
>
> We note that the goal of both Setting 1 and 2 is to provide a way to understand how the number of permissible models in the Rashomon set blows up with small increases in the total number of strategies in simple, constrained settings. Relaxing these assumptions would lead to an even greater blow-up in the number of permissible models. For instance, if one were to allow trades to occur but also allow $\epsilon$-permissability, the number of permissible models would be a lot greater than those presented by Settings 1 and 2.
>
> ---
>
> We will be happy to answer any additional questions or concerns the reviewer may have.

---

> > ### Author Response · Authors · 2025-11-24
> >
> > Dear Reviewer B3Vw,
> >
> > Kindly let us know if our rebuttal has addressed your concerns or if you have any additional concerns regarding our work.
> >
> > We thank you for your time and consideration,
> >
> > Authors

---

### Meta-Review · Area_Chair_xQjd · 2026-01-01

**Summary:**

This paper identified a phenomenon of predictive multiplicity and stated that LLMs do not make complete use of the math reasoning data used to train them. The scores have large diversity. There were also some concerns of LLM generated reviews. One reviewer agreed that LLM was used to polish the wording but the opinions were their own. I did not find definite evidence to prove or disprove that the flagged reviews are LLM generated. However, I did not find the concerns raised in those reviews completely unreasonable.

In summary, there are concerns from multiple reviewers that the title and claims of the paper are a bit misleading. The phenomenon observed by the paper may not be directly implying that the models are not making complete use of the data (and the claim is also not very clearly defined). Moreover, it is unclear how this concept is related to previous known phenomenon that deep learning could converge to different local minima, which are essentially different solutions. Finally, there were concerns of limited proposals that could effectively mitigate the issue, and some obvious baseline mitigations that are directly related to the phenomenon are not tried, such as checkpoint ensembling across data-subset steps, EMA over training, mixture-of-checkpoints.

**Reviewer Concerns:**

Many minor concerns are mostly addressed by the rebuttals. The authors also clarified the relations to catastrophic forgetting and overfitting, which partially addressed the concern of relation to existing known phenomenon. The author acknowledge and clarified the claim of "fully using data", but major change of the narration and possible change of title may be needed. The concern of not testing many potential mitigation baselines are not addressed very satisfactorily.

**Reviewer Scores:**

I expect reviewer B3Vw to keep their rating as the concern of relation to local minima in non-convex optimization is not fully addressed. Actually, the rebuttal did not discuss the three weakness listed by the reviewer.

I expect reviewer AhtP to keep their rating or only slightly increase it as their major concern of missing studies of mitigation baselines are not addressed satisfactorily.

---

### Decision · Program_Chairs · 2026-01-26

Reject